# Adaptive Laboratory Evolution of *Halomonas bluephagenesis* Enhances Acetate Tolerance and Utilization to Produce Poly(3-hydroxybutyrate)

**DOI:** 10.3390/molecules27093022

**Published:** 2022-05-08

**Authors:** Jing Zhang, Biao Jin, Jing Fu, Zhiwen Wang, Tao Chen

**Affiliations:** 1Frontier Science Center for Synthetic Biology and Key Laboratory of Systems Bioengineering of Ministry of Education, Tianjin University, Tianjin 300072, China; zhangjing_2019@tju.edu.cn (J.Z.); jinbiao@tju.edu.cn (B.J.); zww@tju.edu.cn (Z.W.); 2School of Chemical Engineering and Technology, Tianjin University, Tianjin 300072, China; 3Department of Biology and Biological Engineering, Chalmers University of Technology, 412 96 Gothenburg, Sweden; jingf@chalmers.se

**Keywords:** acetate, *Halomonas bluephagenesis*, adaptive laboratory evolution, PHB

## Abstract

Acetate is a promising economical and sustainable carbon source for bioproduction, but it is also a known cell-growth inhibitor. In this study, adaptive laboratory evolution (ALE) with acetate as selective pressure was applied to *Halomonas bluephagenesis* TD1.0, a fast-growing and contamination-resistant halophilic bacterium that naturally accumulates poly(3-hydroxybutyrate) (PHB). After 71 transfers, the evolved strain, B71, was isolated, which not only showed better fitness (in terms of tolerance and utilization rate) to high concentrations of acetate but also produced a higher PHB titer compared with the parental strain TD1.0. Subsequently, overexpression of acetyl-CoA synthetase (ACS) in B71 resulted in a further increase in acetate utilization but a decrease in PHB production. Through whole-genome resequencing, it was speculated that genetic mutations (single-nucleotide variation (SNV) in *phaB*, *mdh,* and the upstream of *OmpA,* and insertion of *TolA*) in B71 might contribute to its improved acetate adaptability and PHB production. Finally, in a 5 L bioreactor with intermittent feeding of acetic acid, B71 was able to produce 49.79 g/L PHB and 70.01 g/L dry cell mass, which were 147.2% and 82.32% higher than those of TD1.0, respectively. These results highlight that ALE provides a reliable method to harness *H. bluephagenesis* to metabolize acetate for the production of PHB or other high-value chemicals more efficiently.

## 1. Introduction

Microbial fermentation is highly promising as it can convert renewable carbon substrates (mainly sugars from corn) to various valuable bioproducts, such as chemicals, materials, and biofuels [1]. The utilization of abundant and low-cost feedstocks is particularly important for achieving the economical and sustainable production of bioproducts [2]. Acetate is an abundant and low-cost feedstock since it can be obtained easily in multiple cheap ways, such as via deacetylation of hemicellulose or chemical synthesis (methanol carbonylation, ethylene oxidation, and alkane oxidation) [2]. The annual production of acetate is around 12.9 million tons, while its price ($350–450 per ton) is already lower than that of conventional sugar ($500 per ton for glucose) in the United States by the year 2018 [3]. More importantly, acetate also has many advantages, such as: (1) acetate is highly soluble in water, allowing easy mass transfer to culture broth and a very low dilution bioreactor volume; (2) microbial acetate consumption raises medium pH, and, thus, the dual effect of carbon feeding and pH control can be realized by using acetic acid as a titrant; (3) acetate is a nonreducing compound, so it does not undergo browning (Maillard) reactions that affect product quality; (4) most industrially relevant microorganisms can naturally metabolize acetate and yield acetyl-CoA, an important starting precursor for the production of value-added bioproducts [1]. Thus, acetate could be a promising carbon source for microbial fermentation processes.

To date, some microorganisms, including *Escherichia coli*, *Saccharomyces cerevisiae*, *Corynebacterium glutamicum*, *Cryptococcus curvatus*, and *Clostridium thermobutyricum*, have been developed by using acetate as the carbon source to produce desired bioproducts, such as succinate, ethanol, itaconic acid, 3-hydroxypropionic acid, free fatty acids, and recombinant protein [2,4,5,6,7,8,9,10,11]. However, most of these products from acetate have a rather low titer and yield compared to conventional carbon sources, such as glucose and glycerol. In most cases, it is suggested that the growth of microorganisms was inhibited or the utilization of acetate was not efficient. Therefore, a chassis with high acetate tolerance and efficient acetate utilization needs to be developed for realizing the blueprint of acetate as a next-generation platform substrate in future industrial biotechnology.

ALE is an efficient strategy to improve substrate utilization and strain robustness (tolerance to high substrate concentrations or toxic compounds), as well as the product titer, yield, and productivity [12]. It has been reported that many microorganisms have applied ALE to improve acetate tolerance, such as *E. coli* and *S. cerevisiae* [13,14,15]. The best-evolved strain tolerated no more than 40 g/L sodium acetate [14] as even small amounts (~0.5 g/L sodium acetate) usually significantly retard the growth of wild-type strains [16]. Hence, it will be more effective to select microorganisms that are more resistant to acetate than the above chassis for ALE.

The next-generation industrial biotechnology (NGIB) chassis *Halomonas bluephagenesis*, a salt-loving bacterium that thrives in high-salt and high-alkali conditions, is a natural poly(3-hydroxybutyrate) (PHB) producer. PHB, a class of biopolymers, is biodegradable and biocompatible in nature and can be used frequently in disposable packages, agriculture, pharmaceuticals, and medical devices, etc. [17]. In addition, *Halomonas bluephagenesis* has also been engineered to produce other metabolic targets, such as itaconate, L-lysine, and cadaverine [18,19,20,21]. The previous study showed that *Halomonas bluephagenesis* is intrinsically resistant to a very high concentration of acetate and can grow well even at 75 g/L sodium acetate [22], which is comparable to well-known acetic acid bacteria (AAB), such as *Acetobacter aceti* and *Gluconobacter suboxydans* (tolerance up to 70 g/L sodium acetate) [23]. However, the acetate utilization ability of *H. bluephagenesis* was still inferior to that of glucose, the dominant carbon source [22].

Hence, in this study, a strategy of ALE was designed to improve the acetate adaptability of *H. bluephagenesis.* The acetate tolerance, growth profile, acetate utilization rate, and PHB production of the parental strain and the evolved strain were then compared. Furthermore, a whole-genome resequencing analysis of the evolved strain was performed to understand what mechanism might be responsible for the improved phenotype.

## 2. Results and Discussion

### 2.1. Adaptive Laboratory Evolution of H. Bluephagenesis in the Presence of Acetate

For adaptive evolution, *H. bluephagenesis* TD1.0 was continuously cultured in the 60 MMA medium. The initial acetate concentration was 20 g/L, and then gradually increased to 120 g/L (the growth of wild-type strain was severely inhibited) [22]. The procedure of ALE for TD1.0 is depicted in Figure 1. After 71 serial transfers, the ALE of TD1.0 was stopped because the cell growth rate did not increase significantly. The endpoint evolved populations were grown on 60 MMA agar plates containing 120 g/L sodium acetate, and a single colony was isolated from that plate. The colony exhibited higher cell density (OD_600_ = 15.62) in 60 MMA medium containing 100 g/L sodium acetate than that of parental strain TD1.0 in 75 g/L sodium acetate (OD_600_ = 10.57) (Figure 2), which suggested that its tolerance to high concentrations of acetate was significantly improved. We thus selected this colony, named B71, for subsequent experiments.

To explore the differences in acetate consumption, cell growth, and PHB production between B71 and TD1.0, fermentation processes were performed using 60 MMA medium containing 27.3 g/L sodium acetate. As shown in Figure 3a, B71 could deplete acetate within 24 h, while TD1.0 depleted the substrate at 36 h, indicating that the acetate utilization rate of B71 (1.14 g/L/h) was increased by 50% compared with that of TD1.0 (0.76 g/L/h). In addition, the growth of B71 was also better than that of TD1.0. After 30 h of cultivation, B71 could reach a maximum OD_600_ of 35.3, whereas the OD_600_ of TD1.0 was only 31.0 when cultured for 42 h. Notably, the intracellular PHB production by B71 was also significantly improved. As shown in Figure 3b, the maximum PHB titer of B71 was 7.27 g/L at 30 h, while the titer of TD1.0 was 4.15 g/L at the same time, and its maximum titer was only 5.86 g/L at 42 h. The PHB production of B71 increased by 24.06% compared with TD1.0.

Combined, these results demonstrate that ALE increases not only the adaptability and utilization of B71 to acetate but also its intracellular PHB production. As is known, PHB is synthesized from acetyl-CoA as the precursor [24], which can be converted by microorganisms from acetate [1]. Hence, improving the acetate tolerance and utilization is helpful to increase acetyl-CoA pools, which, in turn, increases the production of acetyl-CoA-derived compounds. In a recent study, Liu et al. showed that the acetate-adaptive strains *Zymomonas mobilis* ZMA-142 and ZMA-167 exhibited an increase in ethanol yield by 32.21% and 21.16%, respectively [25].

### 2.2. Overexpression of Acetyl-CoA Synthase in B71

In most microorganisms, acetate can be metabolized via two routes catalyzed by acetyl-CoA synthetase (ACS, encoded by *acs*) or acetate kinase/phosphotransacetylase (AckA-Pta, encoded by *ackA* and *pta*, respectively) [3]. Overexpression of *acs* or *ackA*-*pta* is often effective in promoting acetate assimilation, cell growth, and metabolite production [26]. Therefore, in this study, the *acs* gene derived from *Bacillus subtilis* was codon-optimized and cloned into plasmid pN59 equipped with an inducible T7-like promoter P*_Mmp1_* and an intensive RBS, yielding pN59-P*_Mmp1_*-ACS. Subsequently, pN59-P*_Mmp1_*-ACS was conjugated into B71, generating B71-ACS.

Fermentation experiments were performed. As shown in Figure 4a, the growth of B71-ACS was slightly slower than that of B71 in the first 9 h and significantly faster than that of B71 after that. The acetate consumption of B71-ACS was consistent with its growth trend, showing a higher acetate utilization rate during the period of 6 to 18 h. The slowing growth of B71-ACS during the early stage might be related to the metabolic burden caused by plasmid replication and ACS protein expression. Meanwhile, the mature ACS proteins afford a fast acetate utilization rate to generate sufficient acetyl-CoA, a key node in metabolism, thereby improving cell growth and metabolism. In addition, the cell dry weight (CDW) and PHB production were also investigated. B71-ACS achieved the maximum CDW (11.87 g/L) at 30 h, which was slightly higher than that of B71 (11.36 g/L), but its intracellular PHB ratio was only 0.54 g/g (PHB/CDW) (Figure 4b), resulting in a PHB titer of 6.41 g/L, which was lower than that of B71 (7.27 g/L) (Figure 4c). These results demonstrate that ACS overexpression indeed increases acetate metabolism, but this synergistic effect only promoted cell growth rather than PHB production, probably due to the insufficient PHB synthetic flux. Therefore, applying metabolic engineering or synthetic biology approaches to optimize the expression of key enzymes for PHB synthesis, or to decouple cell growth from PHB accumulation, may be beneficial to direct more metabolic flux to PHB synthesis in further study.

### 2.3. Whole-Genome Sequence of the Evolved Strain B71

The accumulation of beneficial mutations during ALE is responsible for the adaptation of strains [27]. To pinpoint the causal mutations that improved acetate fitness, we performed the whole-genome resequencing of the parental strain TD1.0 and the evolved strain B71. Omitting the background mutations found in the parental strain TD1.0 (genetic sequence in conflict with the reference genome NZ_OV350343.1), a total of 48 new genomic mutations were found in B71 compared to TD1.0, and the mutations with gene annotation were listed in Appendix A.

Some of the notable single nucleotide variations (SNVs), including *phaB* (acetoacetyl-CoA reductase encoding gene) and *mdh* (malate dehydrogenase encoding gene), were found in B71 (Table 1). Acetoacetyl-CoA reductase, encoded by *phaB*, is a key rate-limiting enzyme in PHB synthesis, catalyzing the conversion of acetoacetyl-CoA to 3-hydroxybutyryl-CoA [28]. Previously, random mutagenesis of phaB was an effective means to obtain the activity-enhanced mutants that can increase PHB production. Matsumoto et al. obtained two novel activity-enhanced phaB mutants bearing Gln47Leu (Q47L) and Thr173Ser (T173S) substitutions by error-prone PCR. The k_cat_ values of the two mutants were 2.4-fold and 3.5-fold higher than that of the wild-type enzyme, respectively. The PHB content of the recombinant *C. glutamicum* harboring the two mutants was 3-fold and 7-fold higher than that of the wild-type enzyme-containing strain, respectively [29]. In this study, G to C (Ala26Ala) and G to T (Gly29Cys) mutations occurred in the phaB of B71, and the nonsynonymous mutation of Gly to Cys may contribute to increasing the stability of phaB, thereby providing more 3-hydroxybutyryl-CoA for PHB production. It has been demonstrated that glycine is a known secondary structure breaker due to its inability to protect the backbone of the hydrogen bond [30]. In contrast, cysteine behaves as a strong hydrophobic amino acid that stabilizes protein compared to glycine [31]. Thus, the replacement of glycine with cysteine in phaB may increase its hydrophobic effect, thus weakening the penetration of water molecules, which can make the secondary structure more stable. Next, reverse engineering should be considered to verify the function of the phaB mutant (Gly29Cys) in further research.

Malate dehydrogenase (MDH), encoded by *mdh*, is a key enzyme in the tricarboxylic acid (TCA) cycle, catalyzing the production of oxaloacetate from malate [33]. Overoxidation of acetate through the TCA cycle is the main mechanism by which AAB strains detoxify the intracellular acetic acid, and, thus, enhancing the TCA cycle flux by increasing the activity of the related enzymes may improve acetic acid tolerance [34,35]. Based on these notions, we speculated that the MDH mutant (V14A) of B71 might play an important role in upregulating the TCA cycle by supplying more oxaloacetate, resulting in a faster acetate utilization rate and a stronger ability to detoxify acetate than that of TD1.0.

The microbial membrane acts as a protective barrier and thus plays an important role in tolerance to stressful conditions [36]. Changing membrane fluidity and stability is one of the mechanisms by which microbes improve acetate tolerance [37]. Similarly, the adjustment of the membrane structure might also occur in the acetate-evolved strain B71. Mutations in genes relevant to the membrane composition were identified, such as multiple site mutations in the upstream of OmpA family proteins and loss-of-function mutation of the cell integrity protein TolA (Table 1).

Taken together, we speculated that there may be two reasons for improving the acetate fitness of B71: (1) the enhancement of the TCA cycle and PHB synthesis flux accelerates the intracellular acetate metabolism, which can quickly relieve the toxicity of acetate; (2) the change in the cell membrane composition may increase its acetate tolerance. The specific reasons and mechanisms need to be further explored.

Mutations occur randomly during the ALE process, and, when some particular genetic mutations are beneficial for improved fitness or survival, those mutations are naturally selected [38]. For acetate tolerance, some SNVs, including *cspC* (stress protein), *patZ* (peptidyl-lysine N-acetyltransferase) [13], and *rpoA* (the α subunit of the RNA polymerase core enzyme) [39], were also demonstrated to be related to efficient utilization of acetate and tolerance to acetate in the previous studies. Moreover, a chromosomal deletion of 27.3 kb related to the respiration of nitrate (*narP*, *napF*, *napD*, *napA*, *napB*, *napC*), repair of alkylated DNA (*alkB*, *ada*), the *ompC* coding for porin C, the synthesis of cytochromes C (*ccmH*, *ccmD*, *ccmC*, *ccmB*, *ccmA*), thiamine (*apbE*), and colonic acid (*rscD*, *rscB*, *rscC*) was identified in acetate-tolerant strain *E. coli* MS04, contributing to increasing the acetate tolerance [14]. All these mutations were not found in the evolved strain B71, and the role of other mutations (Appendix A) on the phenotype of this strain need to be investigated in further study.

### 2.4. Fed-Batch for Evolved Strain B71

Ultimately, we chose B71 and TD1.0 for fed-batch fermentation in a 5 L bioreactor. For the parental strain TD1.0, the total concentration of sodium acetate supplemented during the entire fermentation process was 112 g/L, and only 93.70 g/L was consumed. After 150 h, TD1.0 produced a maximum CDW of 38.40 g/L containing 52.43% PHB (equivalent to 20.13 g/L PHB) and with a PHB yield of 0.29 g/g acetic acid (Figure 5a). In comparison, the evolved strain B71 exhibited more advantages in these aspects, such as acetate tolerance, acetate consumption capacity, and PHB production. A total of 217 g/L sodium acetate was added, and 199.70 g/L was consumed. The average utilization rate of acetate of B71 was 1.33 g/L/h, which was 1.15 times higher than that of TD1.0 (0.62 g/L/h). After 150 h, B71 produced a maximum CDW of 70.01 g/L containing 71.10% PHB (equivalent to 49.78 g/L PHB) and with a PHB yield of 0.34 g/g acetic acid (Figure 5b). The CDW, PHB titer, and PHB yield of B71 increased by 82.32%, 147.2%, and 17.24%, respectively, compared with TD1.0. In addition, we noticed that B71 exhibited a longer-lasting, more stable, and stronger acetate utilization capacity compared with TD1.0 throughout the fermentation process. These results showed that B71 exhibited the superior performance of high cell density fermentation and high PHB production using acetate as the carbon source. In recent years, reports on the study of using acetate as the carbon source to produce PHB have emerged in an endless stream. Compared with *Yarrowia lipolytica* [40], *Methylorubrum extorquens* [41], *Bacillus cereus* [42], *Aeromonas hydrophilia* [43], and *E. coli* [44], etc., the ability of B71 to convert acetate to PHB was much higher (Table 2). In the future, the optimization of the fermentation conditions and the intervention of metabolic engineering can further improve its ability to metabolize acetate to generate PHB or other value-added chemicals.

## 3. Materials and Methods

### 3.1. Strains and Growth Conditions

Strains and plasmids are listed in Table 3. *E. coli* S17-1 was used as the host for plasmid construction and the donor for conjugation. *H. bluephagenesis* TD1.0 used for ALE was kindly donated by Professor Guo-Qiang Chen from Tsinghua University. For cell growth, the LB medium containing 10 g/L tryptone, 5 g/L yeast extract, and 10 g/L NaCl was used for *E. coli*. The 60 LB medium, which was derived from LB medium supplemented with 60 g/L NaCl, was used for *H. bluephagenesis* strains. 20 g/L agar was added before autoclaving for preparing solid media. 60 MM minimal medium [45] with acetate as the sole carbon source (named 60 MMA) was used for ALE and fermentation processes. The components of 60 MMA medium contain 60 g/L NaCl, 1 g/L yeast extract, 0.2 g/L MgSO_4_·7H_2_O, 9.65 g/L Na_2_HPO_4_·12H_2_O, 1.5 g/L KH_2_PO_4_, 10 mL/L trace element solution I, and 1 mL/L trace element solution II, supplemented with various concentrations of sodium acetate. Trace element solution I comprises 5 g/L Fe (III)-NH_4_-citrate and 2 g/L CaCl_2_ dissolved in 1 M HCl; trace element solution II contains 0.1 g/L ZnSO_4_·7H_2_O, 0.03 g/L MnCl_2_·4H_2_O, 0.3 g/L H_3_BO_3_, 0.2 g/L CoCl_2_·6H_2_O, 0.03 g/L NaMoO_4_·2H_2_O, 0.02 g/L NiCl_2_·6H_2_O, and 0.01 g/L CuSO_4_·5H_2_O. The pH of the 60 MMA medium was adjusted to 8.5–9.0 using 5 M NaOH. Chloramphenicol (25 μg/mL) was added to the above media if needed. Strains used in this study were cultured at 37 °C.

### 3.2. ALE Procedure for H. Bluephagenesis

Evolution was carried out at 37 °C using 50 mL of 60 MMA in a 500 mL Erlenmeyer flask. Specifically, a single colony was inoculated in 5 mL of 60 LB medium for 12 h at 37 °C and 200 rpm to acquire the first seed culture, which was further transferred into 20 mL of 60 LB medium at a volume ratio of 1% for the secondary seed culture preparation. After incubation for 10–12 h, the secondary seed culture was inoculated into 50 mL of the 60 MMA (initially containing 20 g/L sodium acetate) medium at a volume ratio of 2% and cultivated for 12 h at 37 °C and 200 rpm, and followed by being retransferred to the same newly prepared medium at a 2% inoculum volume for several times. When the cell density (OD_600_) no longer increased under 20 g/L sodium acetate, the evolutionary pressure was successively increased to 70 g/L, 80 g/L, and 120 g/L sodium acetate, respectively, repeating the transfer procedure described above for each pressure condition. Finally, a total of 71 consecutive transfers were carried out during the evolution process, as shown in Figure 1. Throughout the evolution process, the acetate concentration was gradually increased from 20 g/L to 120 g/L to strengthen the evolutionary pressure. The resulting culture was plated onto 60 MMA agar plate supplemented with 120 g/L sodium acetate and incubated at 37 °C. A single colony that grew faster, named B71, was obtained and selected for further analyses.

### 3.3. Plasmids Construction and Conjugation

Plasmids used in this study are listed in Table 3. Molecular cloning experiments were carried out according to manufacturers’ instructions or standard procedures. To construct plasmid pN59-P*_Mmp1_*-ACS, the encoding sequence of ACS (acetyl-CoA synthase) from *Bacillus subtilis* was synthesized by Genewiz Biotech Co., Ltd. (Suzhou, China) with codon optimization for *H. bluephagenesis*, and cloned into the high-copy plasmid backbone pN59 (also named pSEVA341). Inducible promoter P_T7-like_ Mmp1 and strong RBS were used to efficiently regulate the expression of *acs*. The DNA sequences were confirmed by sequencing.

The constructed plasmid pN59-P*_Mmp1_*-ACS was transformed into *E. coli* S17-1 using electroporation, and then transformed into *H. bluephagenesis* via conjugation. Specifically, the recombinant *E. coli* S17-1 and *H. bluephagenesis* were separately cultured overnight. Subsequently, cells of *E. coli* S17-1 and *H. bluephagenesis* were harvested by centrifugation at 4000 rpm for 5 min and washed twice using 20 LB medium (LB medium supplemented with 20 g/L NaCl), respectively. Then, the collected cells were mixed at a ratio of 1:1 and incubated on 20 LB plate at 37 °C for 6–8 h. Finally, conjugated bacterial lawn suspended with 60 LB was spread on 60 LB agar plate with relevant antibiotics for 36–48 h incubation at 37 °C to obtain positive colonies.

### 3.4. Shake-Flask and Fed-Batch Fermentation Studies

The 60 MMA medium was used for shake-flask and fed-batch fermentation. For shake-flask studies, single colonies were inoculated in 5 mL of 60 LB medium for 12 h at 37 °C and 200 rpm to acquire the first seed culture, which was further transferred into 20 mL of 60 LB medium at a volume ratio of 1% for the secondary seed culture preparation. After incubation for 10 h, the resulting cultures were inoculated into 500 mL conical flask containing 50 mL of 60 MMA medium at a volume ratio of 5% and cultivated for 48 h at 37 °C and 200 rpm. 1 mM isopropyl-*β*-D-1-thiogalactopyranoside (IPTG) was added when cell density (OD_600_) reached 0.4–0.6 to induce ACS expression. Chloramphenicol (25 μg/mL) was added when necessary.

The seed culture used for fed-batch fermentation was prepared in the same way as that in the shake-flask. A 200 mL seed culture was inoculated into a 5 L bioreactor (Bailun Bio, Shanghai, China) to a working volume of 2 L. 60 MMA medium supplemented with 2.5 g/L yeast extract and 30 g/L sodium acetate was used as an initial medium. The growth temperature was 37 °C, maintained via a cooling circulation pump (Henan Jinghua Instrument, Zhengzhou, China). The pH was maintained at 9.0 by adding 5 M NaOH using an automatic pump. Meanwhile, the dissolved oxygen (DO) level was maintained at ~30% of air saturation by regulating the airflow rate (a maximum flow rate of 1 vvm) and the stirring rate (200–800 r/min). During fermentation, pure acetic acid was fed intermittently to prevent depletion of acetate in the medium. 5 g/L NH_4_Cl was added at 15 h after the beginning of the fermentation. The evaporation volume during fermentation was corrected with sterile water. Samples were taken every 4–6 h and the cell dry weight (CDW), acetate concentration, and PHB content were determined.

### 3.5. Analytical Methods

Cells were collected from 5 mL cell cultures by centrifuging at 5000 rpm for 5 min, and washed twice with distilled water. The CDW was calculated by measuring the mass of harvested cells after lyophilization for 12 h. PHB content was determined by a modification of the Sun et al. methodologies [49]. Briefly, 30–40 mg of lyophilized cell samples were mixed with 2 mL of chloroform and 2 mL of esterification solution (containing 97% (*v*/*v*) chromatographic grade methanol, 3% (*v*/*v*) concentrated sulfuric acid, and 1 g/L benzoic acid) in a screw-capped tube. Methanolysis was carried out at 100 °C for 4 h. After cooling to room temperature, 1 mL of deionized water was added and vortexed for 1 min. The resulting mixture was allowed to separate into layers, and the lower organic phase was used for gas chromatography (GC) analysis [50]. GC analysis was performed using a Scion 456-GC (Bruker, Germany) equipped with an HP-5 capillary column (30 m × 0.32 mm, Agilent, Santa Clara, CA, USA). Sample (1 μL) in the organic phase was injected with nitrogen as the carrier gas, and the oven temperature was programmed at 80 °C for 1.5 min, then increased at 30 °C/min to 140 °C, then 40 °C/min to 240 °C and run for 2 min. The injector temperature was 240 °C, and the flame ionization detector (FID) temperature was 250 °C. PHB purchased from Sigma-Aldrich (St. Louis, MO, USA) was used as the standard.

Concentrations of acetate in the medium were determined via high-performance liquid chromatography (HPLC). An Agilent 1100 Series HPLC system (Agilent Technologies, Waldbronn, Germany) equipped with an Aminex HPX-87H column (300 × 7.8 mm, 9 μm, Bio-Rad, Hercules, CA, USA) and a refractive index detector were employed. The mobile phase was 5 mM H_2_SO_4_ with a flow rate of 0.4 mL/min. The column temperature was maintained at 65 °C.

### 3.6. Whole-Genome Sequencing

The strain was cultivated overnight in 50 mL of 60 MMA medium containing 30 g/L sodium acetate at 37 °C and 200 rpm in an incubator shaker. The cells were harvested by centrifugation (5000 rpm, 10 min, 4 °C) and washed twice to obtain the cellular pellet. The whole-genome resequencing was commissioned by Genewiz Biotech Co., Ltd. (Suzhou, China).

## 4. Conclusions

Overall, ALE was first successfully applied to improve the acetate adaptability of *H. bluephagenesis*. As a result, a better-performing strain, B71, was obtained, which exhibited both superior acetate fitness and high PHB production. In the future, B71 has the potential as a robust platform for the production of other high-value chemicals with acetate as a carbon feedstock. Furthermore, the whole-genome resequencing analysis allowed identifying genetic mutations in response to acetate, which can provide direction to construct new *H. bluephagenesis* strains with an improved ability to convert acetate into valuable compounds through genetic engineering.

## Figures and Tables

**Figure 1 molecules-27-03022-f001:**
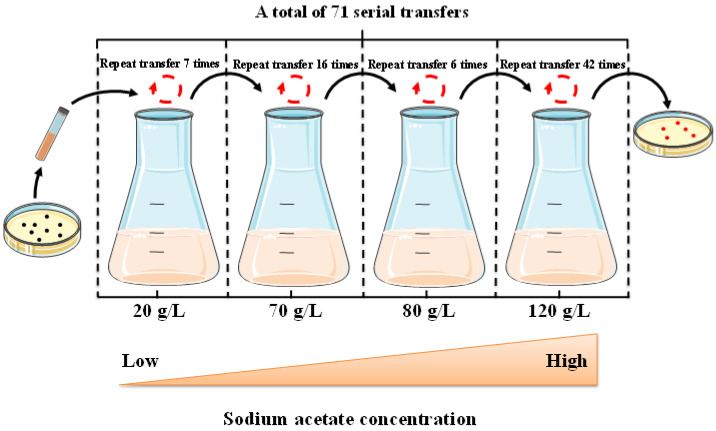
Adapted cultivation of *H. bluephagenesis* TD1.0. The single colony (black spot) was serially transferred in 60 MMA medium with increasing concentrations of sodium acetate, and, finally, evolved colonies (red spot) were isolated on 60 MMA solid plates containing 120 g/L sodium acetate.

**Figure 2 molecules-27-03022-f002:**
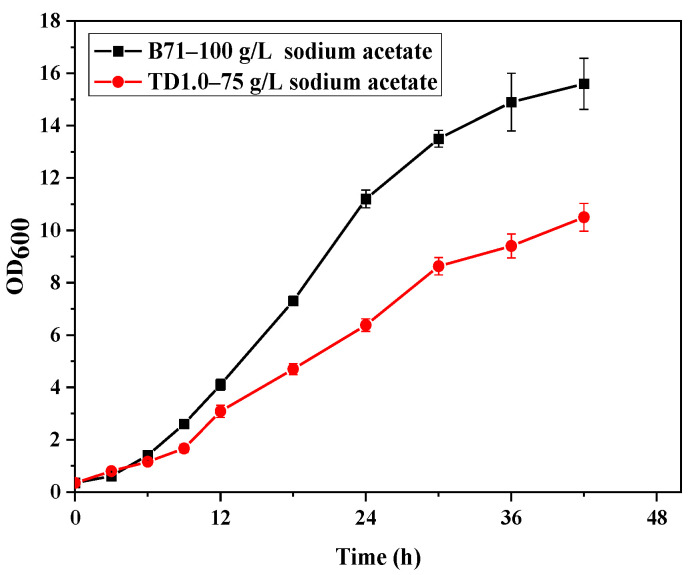
Comparison of acetate tolerance of the evolved strain B71 (black solid square) and the parental strain TD1.0 (red solid circle). B71 and TD1.0 were grown in 60 MMA minimal medium containing 100 g/L and 75 g/L sodium acetate, respectively. Error bars represent standard deviation (SD), *n* = 3.

**Figure 3 molecules-27-03022-f003:**
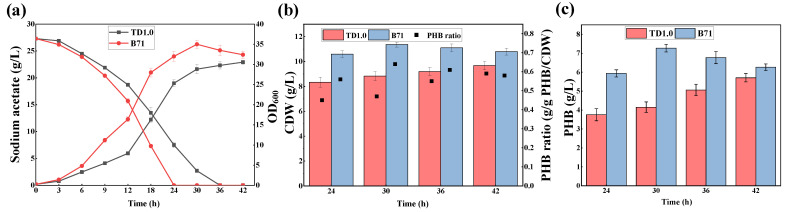
Comparison of growth, acetate utilization, and PHB production of TD1.0 and B71. Strains were grown in 60 MMA minimal medium containing 27.3 g/L sodium acetate. (**a**) Acetate concentration and OD_600_. (**b**) CDW and PHB ratio. (**c**) PHB production. Error bars represent standard deviation (SD), *n* = 3.

**Figure 4 molecules-27-03022-f004:**
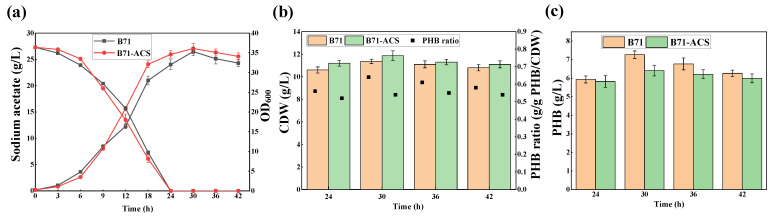
Comparison of growth, acetate utilization, and PHB production of B71 and B71-ACS. B71 and B71-ACS were grown in 60 MMA minimal medium containing 27.3 g/L sodium acetate. (**a**) Acetate concentration and OD_600_. (**b**) CDW and PHB ratio. (**c**) PHB production. Error bars represent standard deviation (SD), *n* = 3.

**Figure 5 molecules-27-03022-f005:**
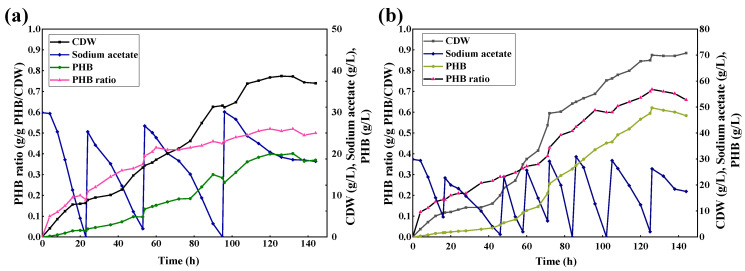
Fed-batch fermentation of (**a**) TD1.0 and (**b**) B71 with acetate as the carbon source in 5 L bioreactor.

**Table 1 molecules-27-03022-t001:** Representative mutations in B71.

Gene	NCBI Number	Function	Position	Type ^a^	Description ^b^
*phaB*	WP_009725067.1	The rate-limiting enzyme for PHB synthesis	Exonic	SNV SNV	G→C,Ala26Ala G→T,Gly29Cys
*mdh*	WP_186004630.1	Key enzyme of TCA cycle	Exonic	SNV	C→T,Val14Ala
*TolA*	WP_050801109.1	Cell envelope integrity protein	Exonic	In	175429: 30 bp
*OmpA* family protein	WP_009724794.1	Protect cells from the effects of environmental stresses and maintain osmotic balance in cells [32]	Upstream Upstream Upstream Upstream Upstream Upstream Upstream	SNV SNV SNV SNV SNV SNV SNV	19715: C→T, 19716: T→G, 19728: G→A, 19731: A→G, 19732: G→A, 20622: A→G, 21072: G→A

^a^ SNV, single-nucleotide variation; In, insertion. ^b^ Presented as nucleotide change: base substitution (if the mutation was in a coding region, the resulting change in amino acid) and base insertion.

**Table 2 molecules-27-03022-t002:** Comparison of PHA production of several microorganisms using acetate as carbon source.

Strain	Carbon Source	PHB (g/L)	CDW (g/L)	Fermentation Mode	Reference
*Y. lipolytica*	Acetate	0.24	6.2	Shake-flasks	[40]
*Y. lipolytica*	Acetate	7.35	72.01	Fed-batch	[40]
*M. extorquens*	Acetate	0.7	2.06	Shake-flasks	[41]
*B. cereus*	Acetate	1.46	2.94	Shake-flasks	[42]
*A. hydrophilia*	Acetate	0.55	2.77	Shake-flasks	[43]
*E. coli* JM109 (pBHR68+pMCS-pta-ackA)	Acetate	1.27	3.02	Shake-flasks	[44]
*H. bluephagenesis* B71	Acetate	7.27	11.36	Shake-flasks	This study
*H. bluephagenesis* B71	Acetate	49.78	70.01	Fed-batch	This study

**Table 3 molecules-27-03022-t003:** Bacterial strains and plasmids used in this study.

Strains/Plasmids	Genotype/Description	Source/Reference
Strains		
*E. coli* S17-1 pir	recA, thi-1. pro, hsdR, RP4-2-Tc::Mu-Km::Tn7, a donor strain used for conjugation	[46]
*H. bluephagenesis* TD1.0	Wild type *H. bluephagenesis* TD01 derivative with Mmp1 RNA polymerase integrated into the genome, P*_J23110_*-lacI-Ptac-MmP1	[47]
B71	Acetate-adapted strain	This study
B71-ACS	B71 harboring pN59-P*_Mmp1_*-ACS	This study
Plasmids		
pN59	A high copy number expression vector, ColE1 replication origin, oriT, CmR	[48]
pN59-P*_Mmp1_*-ACS	pN59 derivates, containing codon-optimized *acs* gene driven by P*_Mmp1_*, a T7-like inducible promoter, CmR	This study

## Data Availability

The data presented in this study are available in Appendix A.

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
