# Peer review of "Adaptive Laboratory Evolution of *Halomonas bluephagenesis* Enhances Acetate Tolerance and Utilization to Produce Poly(3-hydroxybutyrate)"

_molecules, 2022, doi:10.3390/molecules27093022_

Round 1
Reviewer 1 Report
This paper addresses the need to obtain microbial production hosts that are highly tolerant to acetate and utilize it as carbon source. The authors selected the microbe Halomonas bluephagenesis and demonstrated that a strain with increased growth and polyhydroxybutyrate production can be obtained by performing adaptive laboratory evolution. Genome sequencing of the top isolate and heterologous gene expression provided important insights on its acetate metabolism and targets for future improvements. The manuscript is concise and well written, and the goals, methods and conclusions are clear and supported by the data.
Minor comments:
1) The concentration gradient that was used in the ALE experiment is not explicitly mentioned and it would be helpful to specify. There are four concentrations depicted in Fig. 1 but I am assuming that these were included for graphical purposes and a slower gradient was actually used for the 71 culture transfers. Please clarify.
2) Section 3.5 seems to be missing information on the reaction conditions used for esterification of the lyophilized cells, the GC program and detector conditions, and standards used for quantification. Please provide this information.
3) Some adjectives used in the conclusion are in my opinion vague and should be avoided, e.g. "excellent strain", "great potential" and "robust platform".
Author Response
1) The concentration gradient that was used in the ALE experiment is not explicitly mentioned and it would be helpful to specify. There are four concentrations depicted in Fig. 1, but I am assuming that these were included for graphical purposes and a slower gradient was actually used for the 71 culture transfers. Please clarify.
Our response: Thank you for your advice. We have added a description of the ALE process in Section 3.2 to clarify the transfer operation and the use of acetate concentrations in detail. “Specifically, a single colony was inoculated in 5 mL of 60 LB medium for 12 h at 37 °C and 200 rpm to acquire the first seed culture, which was further transferred into 20 mL of 60 LB medium at a volume ratio of 1% for the secondary seed culture preparation. After incubation for 10 – 12 h, the secondary seed culture was inoculated into 50 mL of the 60 MMA (initially containing 20 g/L sodium acetate) medium at a volume ratio of 2% and cultivated for 12 h at 37 °C and 200 rpm, and followed by retransferred to the same new-prepared medium at a 2% inoculum volume for several times. When the cell density (OD600) no longer increased under 20 g/L sodium acetate, the evolutionary pressure was successively increased to 70 g/L, 80 g/L and 120 g/L sodium acetate, respectively, repeating the transfer procedure described above for each pressure condition. Finally, a total of 71 consecutive transfers were carried out during the evolution process, as shown in Figure 1.”
2) Section 3.5 seems to be missing information on the reaction conditions used for esterification of the lyophilized cells, the GC program and detector conditions, and standards used for quantification. Please provide this information.
Our response: Thank you for your advice. We have added the description in Section 3.5. “PHB content was determined by a modification of the Sun et al. methodologies [41]. Briefly, 30 – 40 mg of lyophilized cell samples were mixed with 2 mL of chloroform and 2 mL of esterification solution (containing 97% (v/v) chromatographic grade methanol, 3% (v/v) concentrated sulfuric acid and 1 g/L benzoic acid) in a screw-capped tube. Methanolysis was carried out at 100°C for 4 h. After cooling to room temperature, 1 mL of deionized water was added and vortexed for 1 min. The resulting mixture was left to stand for phase separation, and the organic phase of the lower layer was used for gas chromatography (GC) analysis. GC analysis was performed using a Scion 456-GC (Bruker, Germany) equipped with an HP-5 capillary column (30 m × 0.32 mm, Agilent, USA). Sample (1 μL) in the organic phase was injected with nitrogen as the carrier gas, and the oven temperature was programmed at 80 °C for 1.5 min, then increased at 30 °C/min to 140 °C, then 40 °C/min to 240 °C and run for 2 min. The injector temperature was 240 °C and the flame ionization detector (FID) temperature was 250 °C. PHB purchased from Sigma-Aldrich was used as the standard.”
3) Some adjectives used in the conclusion are in my opinion vague and should be avoided, e.g. "excellent strain", "great potential" and "robust platform".
Our response: Thank you for your advice. We have modified the use of relevant adjectives. We changed "excellent strain" to “better-performed”. We changed "great potential" to "the potential".
Reviewer 2 Report
Comments and Suggestion
Current work Adaptive laboratory evolution of Halomonas bluephagenesis 2 enhances acetate tolerance and utilization to produce 3 poly(3-hydroxybutyrate) is well written and interesting. The manuscript deal with the genetic manipulation for increased production of 3 poly(3-hydroxybutyrate). Manuscript is well after few clarifications the manuscript can be accepted for the publication.
Author should also mention the reason that why after 71 serial transfers the productivity decreased. Is there any medication in the physical factor that may affect the growth and further production.
In the introduction part author should mention the importance of the PHB
Author should include a comparative production efficiency of PHB with the other isolates too, which will highlight the importance and production of PHB with the current isolate.
SNV analysis applied in other species should be included in the discussion.
There is need to include statically section in methods section.

Author Response
1) Author should also mention the reason that why after 71 serial transfers the productivity decreased. Is there any medication in the physical factor that may affect the growth and further production.
Our response: Thank you for your advice. The productivity of B71 obtained by 71 serial transfers was improved compared to that of the parental strain TD.1.0 (Fig. 3). Further overexpression of the plasmid pN59-PMmp1-ACS in B71 resulted in a decrease in productivity, which might be mainly due to the burden of plasmid replication and protein expression affecting cell growth and metabolism, a phenomenon that was also common in other species. The amount of the added drugs (including drug resistance and inducers) was very small, and it is hard to say whether it had negative effects on cell growth and production. It can be compared with the strain overexpressing the empty plasmid (pN59-PMmp1) to verify this guess. The purpose of ACS overexpression was to improve acetate utilization and PHB production. Since acs overexpression using a plasmid system has no positive effect, integration of the acs gene into the genome can be investigated in the future.
2) In the introduction part author should mention the importance of the PHB.
Our response: Thank you for this advice. We have added a description about the importance of PHB in Introduction part. “PHB, a class of biopolymers, is biodegradable and biocompatible in nature and can be used frequently in disposable packages, agriculture, pharmaceuticals, and medical devices, etc. [17].”
3) Author should include a comparative production efficiency of PHB with the other isolates too, which will highlight the importance and production of PHB with the current isolate.
Our response: Thank you for this professional advice. We have reviewed some researches on the synthesis of PHB by other microorganisms using acetate as carbon source in recent years. By comparison, the production capacity of B71 in both shake flasks and fermenters was much higher, which was summarized in Table 2. A comparison of production efficiency of PHB with other species was added in Section 2.4. “In recent years, reports on the study of using acetate as a carbon source to produce PHB have emerged in an endless stream. Compared with Yarrowia lipolytica [38], Methylorubrum extorquens [39], Bacillus cereus [40], Aeromonas hydrophilia [41] and E. coli [42] etc., the ability of B71 to convert acetate to PHB was much higher (Table 2).”
4) SNV analysis applied in other species should be included in the discussion.
Our response: Thank you for your advice. We have added a discussion about this in Section 2.3. “Matsumoto et al. has obtained two novel activity-enhanced phaB mutants bearing Gln47Leu (Q47L) and Thr173Ser (T173S) substitutions by error-prone PCR. The kcat values of the two mutants were 2.4-fold and 3.5-fold higher than that of the wild-type enzyme, respectively. The PHB content of the recombinant C. glutamicum harboring the two mutants was 3-fold and 7-fold higher than that of the wild-type enzyme-containing strain, respectively [29].” “Mutations occur randomly during ALE process, and when some particular genetic mutations are beneficial for improved fitness or survival, those mutations are naturally selected [38]. For acetate tolerance, some SNV including cspC (stress protein), patZ (pep-tidyl-lysine N-acetyltransferase) [13] and rpoA (the α subunit of the RNA polymerase core enzyme) [38] were also demonstrated to be related to efficient utilization of acetate and tolerance to acetate in the previous studies. Also, a chromosomal deletion of 27.3 kb related to the respiration of nitrate (narP, napF, napD, napA, napB, napC), repair of alkylated DNA (alkB, ada), the ompC coding for porin C, the synthesis of cytochromes C (ccmH, ccmD, ccmC, ccmB, ccmA), thiamine (apbE), and colonic acid (rscD, rscB, rscC) was identified in acetate tolerant strain E. coli MS04, contributing to increase the acetate tolerance [14]. All these mutations were not found in the evolved strain B71, and the role of other mutations (Table S1) on phenotype of this strain need to be investigated in further study. ”
5) There is need to include statistically section in methods section.
Our response: Thank you for this professional advice. The data processing in the manuscript was to calculate the mean and standard deviation based on the results of 3 parallel experiments, and the graphs were drawn using Origin 9 software. Statistical analyses were not presented separately in the Materials and methods section because complex statistical methods are not involved.
Reviewer 3 Report
In this paper, adaptive laboratory evolution with sodium acetate was applied to the wild-type strain, Halomonas bluephagenesis TD1.0. After 71 transfers, strain B71 was obtained. It can grow and synthesize P(3HB) at high concentrations of sodium acetate. The concept of the article is interesting and suitable to publish in Molecules. However, it needs to address some comments, and thus require substantial major revision to improve the quality of the manuscript.
- Add information about the wild-type strain, Halomonas bluephagenesis TD1.0 in the 3.1 section. Have you got it from the some Collection of microorganisms?
- Write the composition of the medium 60 MMA
Write in more detail the process of strain adaptation to sodium acetate.
You transferred a single colony to a flask containing 50 ml of medium. Bacteria were cultivated during 12 hours. What was the optical density of the culture and the biomass concentration after 12 hours? The initial concentration of acetate was 20 g/L. How much g/L of acetate did you add in the flask for each subsequent transfer in the new-prepared medium?
- What was the density of the inoculum in the fed-batch fermentation study? What OD and concentration of biomass were in the bioreactor at the beginning of cultivation? Usually cell concentration in the inoculum influence the final biomass yield and polymer content.
- Fig. 5B. How can you explain that during the first 45-50 hours the biomass concentration increased to about 20 g/L, then in the next 20 hours the biomass increased to almost 50 g/L?
- Did you determine the concentration of ammonium chloride in the medium during fed-batch cultivation? Why did you add ammonium chloride? Yeast extract has already been added to the medium. It is also a source of nitrogen. Excess nitrogen is known to reduce PHA accumulation. Did you determine the optimal concentrations of nitrogen sources?
- Add a reference to the method for determining the content of the polymer in biomass
- Add concentration of biomass at the Figure 3.
Author Response
1) Add information about the wild-type strain, Halomonas bluephagenesis TD1.0 in the 3.1 section. Have you got it from the some Collection of microorganisms?
Our response: Thank you for your advice. Halomonas bluephagenesis TD1.0 used in this study was kindly donated by Professor Guo-Qiang Chen from Tsinghua University. We have added the information in Section 3.1.
2) Write the composition of the medium 60 MMA.
Our response: Thank you for your advice. We have added the composition of the 60 MMA medium in Section 3.1. "The components of 60 MMA medium contains 60 g/L NaCl, 1 g/L yeast extract, 0.2 g/L MgSO4·7H2O, 9.65 g/L Na2HPO4·12H2O, 1.5 g/L KH2PO4, 10 mL/L trace element solution I, 1 mL/L trace element solution II, supplemented with various concentrations of sodium acetate. Trace element solution I comprises 5 g/L Fe (III)-NH4-citrate and 2 g/L CaCl2 dissolved in 1 M HCl; Trace element solution II contains 0.1 g/L ZnSO4·7H2O, 0.03 g/L MnCl2·4H2O, 0.3 g/L H3BO3, 0.2 g/L CoCl2·6H2O, 0.03 g/L NaMoO4·2H2O, 0.02 g/L NiCl2·6H2O, and 0.01 g/L CuSO4·5H2O. The pH of the 60 MMA medium was adjusted to 8.5 – 9.0 using 5 M NaOH."
3) Write in more detail the process of strain adaptation to sodium acetate. You transferred a single colony to a flask containing 50 mL of medium. Bacteria were cultivated during 12 hours. What was the optical density of the culture and the biomass concentration after 12 hours? The initial concentration of acetate was 20 g/L. How much g/L of acetate did you add in the flask for each subsequent transfer in the new-prepared medium?
Our response: Before evolution, a single colony was first cultured in 60 LB to obtain seed culture, and then inoculated into 60 MMA medium containing sodium acetate for ALE. We left out the description of the seed culture preparation, which has been added in Section 3.2. Meanwhile, the detailed steps of the ALE process were also supplemented in Section 3.2. “Specifically, a single colony was inoculated in 5 mL of 60 LB medium for 12 h at 37 °C and 200 rpm to acquire the first seed culture, which was further transferred into 20 mL of 60 LB medium at a volume ratio of 1% for the secondary seed culture preparation. After incubation for 10 – 12 h, the secondary seed culture (OD600≈10) was inoculated into 50 mL of the 60 MMA (initially containing 20 g/L sodium acetate) medium at a volume ratio of 2% and cultivated for 12 h at 37 °C and 200 rpm, and followed by retransferred to the same new-prepared medium (containing 20 g/L sodium acetate) at a 2% inoculum volume for several times. When the cell density (OD600) no longer increased under the 20 g/L sodium acetate, the evolutionary pressure was successively increased to 70 g/L, 80 g/L and 120 g/L sodium acetate, respectively, repeating the transfer procedure described above for each pressure condition. Finally, a total of 71 consecutive transfers were carried out during the evolution process, as shown in Figure 1.”
4) What was the density of the inoculum in the fed-batch fermentation study? What OD and concentration of biomass were in the bioreactor at the beginning of cultivation? Usually cell concentration in the inoculum influence the final biomass yield and polymer content.
Our response: The OD600 of the secondary seed culture was about 10, and it was inoculated into the fermenter at a volume ratio of 10%, so the OD600 in the bioreactor at the beginning of cultivation was about 1.
5) Fig. 5B. How can you explain that during the first 45-50 hours the biomass concentration increased to about 20 g/L, then in the next 20 hours the biomass increased to almost 50 g/L?
Our response: The CDW containing PHB was increased to 20 g/L during the first 50 hours ( including 14 g/L biomass without PHB and 6 g/L PHB), and the nitrogen supply was sufficient during this period. In the next 20 hours the CDW containing PHB increased to 48 g/L (including 26 g/L biomass without PHB and 22 g/L PHB). In fact, the biomass without PHB was almost stable from 50 to 124 h because of insufficient nitrogen supply, so the increased CDW during this period was due to intracellular PHB accumulation.
6) Did you determine the concentration of ammonium chloride in the medium during fed-batch cultivation? Why did you add ammonium chloride? Yeast extract has already been added to the medium. It is also a source of nitrogen. Excess nitrogen is known to reduce PHA accumulation. Did you determine the optimal concentrations of nitrogen sources?
Our response: The initial concentration of yeast extract in the fermenter was incorrect, and the actual amount added was 2.5 g/L, which has been corrected in Section 3.4. The nitrogen provided by 2.5 g/L yeast extract was insufficient, thus ammonium chloride was added to ensure that the cells could grow at a high density. We did not measure the concentration of ammonium chloride in the fermenter. The amount of ammonium chloride supplementation was based on the research of the fermentation process in PHB production using Halomonas TD01 reported by Tan et al. [1]. In Tan et al. study, the initial concentrations of the nitrogen sources used in fermentor included 1 g/L yeast extract, 3 g/L NH4Cl and 2 g/L urea, followed by continuous addition of 1 g/L ammonium chloride during the 8 − 24 h fermentation period.
Excessive nitrogen sources are not conducive to PHB production, thus we supplemented ammonium chloride in the early stage of fermentation as Tan et al. did. Only fed acetate in the later stage of fermentation creating a nitrogen-limited condition that favors PHB synthesis. We have not optimized the nitrogen source concentration, which can be considered in future research.
7) Add a reference to the method for determining the content of the polymer in biomass.
Our response: We have added the PHB content determination method and a reference in Section 3.5. “PHB content was determined by a modification of the Sun et al. methodologies [41]. Briefly, 30 – 40 mg of lyophilized cell samples were mixed with 2 mL of chloroform and 2 mL of esterification solution (containing 97% (v/v) chromatographic grade methanol, 3% (v/v) concentrated sulfuric acid and 1 g/L benzoic acid) in a screw-capped tube. Methanolysis was carried out at 100°C for 4 h. After cooling to room temperature, 1 mL of deionized water was added and vortexed for 1 min. The resulting mixture was left to stand for phase separation, and the organic phase of the lower layer was used for gas chromatography (GC) analysis. GC analysis was performed using a Scion 456-GC (Bruker, Germany) equipped with an HP-5 capillary column (30 m × 0.32 mm, Agilent, USA). Sample (1 μL) in the organic phase was injected with nitrogen as the carrier gas, and the oven temperature was programmed at 80 °C for 1.5 min, then increased at 30 °C/min to 140 °C, then 40 °C/min to 240 °C and run for 2 min. The injector temperature was 240 °C and the flame ionization detector (FID) temperature was 250 °C. PHB purchased from Sigma-Aldrich was used as the standard.”
8) Add concentration of biomass at the Figure 3.
Our response: We have added the concentration of biomass (CDW) in Figure 3.
Round 2
Reviewer 3 Report
The paper could be published in the Molecules